# Effects of Spirituality, Knowledge, Attitudes, and Practices toward Anxiety Regarding COVID-19 among the General Population in INDONESIA: A Cross-Sectional Study

**DOI:** 10.3390/jcm9123798

**Published:** 2020-11-24

**Authors:** Yohanes Andy Rias, Yafi Sabila Rosyad, Roselyn Chipojola, Bayu Satria Wiratama, Cikra Ikhda Safitri, Shuen Fu Weng, Chyn Yng Yang, Hsiu Ting Tsai

**Affiliations:** 1School of Nursing, College of Nursing, Taipei Medical University, Taipei 11031, Taiwan; yohanes.andi@iik.ac.id (Y.A.R.); rosechipojola@gmail.com (R.C.); 2Faculty of Health and Medicine, College of Nursing, Institut Ilmu Kesehatan Bhakti Wiyata, Kediri 64114, Indonesia; 3Faculty of Health and Medicine, College of Nursing, Sekolah Tinggi Ilmu Kesehatan Yogyakarta, Yogyakarta 55281, Indonesia; rosyad2yafi@gmail.com; 4Department of Epidemiology and Biostatistics, Faculty of Medicine Public Health and Nursing, Universitas Gadjah Mada, Yogyakarta 55281, Indonesia; bayu.satria@ugm.ac.id; 5Department of Pharmacology, College of Pharmacology, Akademi Farmasi Mitra Sehat Mandiri Sidoarjo, Sidoarjo 61262, Indonesia; cikraikhda@gmail.com; 6Division of Endocrinology and Metabolism, Department of Internal Medicine, Taipei Medical University Hospital, Taipei 11031, Taiwan; sfweng@ntu.edu.tw; 7Division of Endocrinology and Metabolism, Department of Internal Medicine, School of Medicine, College of Medicine, Taipei Medical University, Taipei 11031, Taiwan; 8Integrated Medical Examination Center, Taipei Medical University Hospital, Taipei 11031, Taiwan; caring@h.tmu.edu.tw; 9Post-Baccalaureate Program in Nursing, College of Nursing, Taipei Medical University, Taipei 11031, Taiwan

**Keywords:** anxiety, attitudes, COVID-19, knowledge, practices, spirituality, Indonesia

## Abstract

Background: Currently, the determinants of anxiety and its related factors in the general population affected by COVID-19 are poorly understood. We examined the effects of spirituality, knowledge, attitudes, and practices (KAP) on anxiety regarding COVID-19. Methods: Online cross-sectional data (*n* = 1082) covered 17 provinces. The assessment included the Daily Spiritual Experiences Scale, the Depression, Anxiety, and Stress Scale, and the KAP-COVID-19 questionnaire. Results: Multiple linear regression revealed that individuals who had low levels of spirituality had increased anxiety compared to those with higher levels of spirituality. Individuals had correct knowledge of early symptoms and supportive treatment (K3), and that individuals with chronic diseases and those who were obese or elderly were more likely to be severe cases (K4). However, participants who chose incorrect concerns about there being no need for children and young adults to take measures to prevent COVID-19 (K9) had significantly lower anxiety compared to those who responded with the correct choice. Participants who disagreed about whether society would win the battle against COVID-19 (A1) and successfully control it (A2) were associated with higher anxiety. Those with the practice of attending crowded places (P1) had significantly higher anxiety. Conclusions: Spirituality, knowledge, attitudes, and practice were significantly correlated with anxiety regarding COVID-19 in the general population.

## 1. Introduction

The novel coronavirus disease 2019 (COVID-19) is a group of complex respiratory syndromes caused by a beta-coronavirus called severe acute respiratory syndrome coronavirus 2 (SARS-CoV-2) [1]. COVID-19 is currently fostering a rapidly growing global health disaster [2]. As of 26 October 2020, the World Health Organization (WHO) has estimated that the prevalence of people infected with COVID-19 globally had reached 42,966,344 [3]. In particular, this threat has emerged in Indonesia, where COVID-19 prevalence is estimated to be around 394,454 with a mortality rate of 3.4% [4]. Currently, the struggle against COVID-19 is still ongoing in Indonesia [5]. The rising menace of the pandemic has led to a global climate of mental health concerns due to social distancing, travel restrictions, and rumors spreading misinformation [6,7,8]. Unfortunately, mental health problems may trigger a series of physiological events and reduce immunity, and good mental health is vital in curbing infectious transmission [9,10,11].

A recent study in China revealed that vicarious traumatization scores of the general population were significantly more likely to be higher than those of front-line medical professionals [12]. Consequently, these situations urgently need guidelines and/or public health authorities to promote crisis management and safeguard the mental health of the population [13,14]. More than 80% of the general population is aware that mental health concerns need to be addressed [15]. Approximately 8.1%, 16.5%, and 28.8%, of the general Chinese population, respectively have stress, depression, and anxiety symptoms due to the COVID-19 pandemic [6]. Another study from Turkey observed that 115 (45.1%) people were experiencing feelings of anxiety [16]. Moreover, an extensive study determined that 2053 (19.1%) people of an Iranian population exhibited severe symptoms of anxiety, 2291 (21.3%) of them had moderate symptoms, and 1128 (10.5%) had mild symptoms [17]. When anxiety levels increase, individuals may become vague and irrational when reacting to the COVID-19 pandemic [6,18,19,20]. In contrast, individuals with a low level of anxiety are associated with a highly relaxed manner, which consequently improves self-control and encourages adherence to personal protective health practices [19]. However, no epidemiological research has investigated the correlation between anxiety and the COVID-19 pandemic in Indonesia. Thus, a study of the threat of anxiety in Indonesia should immediately be conducted.

Knowledge, attitudes, and practices (KAP) towards COVID-19 play major roles in assessing the willingness of a community to adopt behavioral change initiatives during the pandemic [21,22]. KAP empirical studies can reveal fundamental information to determine the types of intervention which can effectively curb an infectious disease. Consequently, improving KAP is a potentially valuable strategy for better insight into addressing misconceptions [22,23,24]. Moreover, good levels of knowledge were positively associated with optimistic attitudes and appropriate practices [21,25]. Meanwhile, lessons learned from the 2003 SARS outbreak showed that knowledge and practice towards infectious diseases were significantly related to a low level of anxiety [26], which might further complicate efforts to prevent the spread of a disease. However, levels of anxiety in India and China were high, even though the general population of India [15] and China [11] had a reasonably good level of knowledge and attitudes. Moreover, KAP and their associations with amelioration of anxiety have not been clarified in Indonesia. As Indonesia is currently experiencing difficult impacts due to the COVID-19 pandemic [5], relationships between KAP and the level of anxiety need to be clarified.

Spirituality as a complementary treatment in health care is a key factor in reducing psychological outcomes, especially anxiety [27,28]. Generally speaking, spirituality and religiosity are often used interchangeably in studies [27,28,29,30]. Spirituality can contribute to reduce anxiety and support health protection by the power of faith [29,30]. Indonesia is unique because most of the population has a higher level of positive spirituality related to health conditions [31,32]. Improving the level of spirituality is potentially a valuable strategy to restrain mental illness [32], psychological impacts, post-traumatic stress disorder, and anxiety [33]. However, personal spirituality may become the source of host resistance and/or resilience [34]. Therefore, a reasonably high level of spirituality may be associated with reduced or increased anxiety. However, such relationships require clarification, especially in Indonesia.

Previous studies have showed that COVID-19 causes psychological problems [6,8,11,14,15]. Thus, survey findings are of great practical significance to all levels of government for the provision of KAP and spirituality to reduce high levels of anxiety. Therefore, this study assessed the prevalence of anxiety and identified positively and negatively associated factors contributing to anxiety in Indonesia as a unique part of the world.

## 2. Methods

### 2.1. Study Design and Population

Primary data were collected using a community-based cross-sectional study design to select members of the general population in 17 provinces from western, central, and eastern regions of Indonesia. Convenience sampling was carried out by distributing an online survey through a Google Form link via WhatsApp, Instagram, Facebook, and Twitter, which are the most popular and accessible social media platforms in Indonesia. While Instagram and Twitter are more popular among the younger generation, Facebook is generally preferred by older Indonesians. We utilized different approaches to targeting as many respondents as possible from across the region during the 7 April–30 May 2020 data collection period. This involves relying on researchers’ technical and personal networks and engaging with and sharing the survey through social media influencers and community lenders. The inclusion criteria to fill in the Google form were: Indonesian civilian, adult aged 17~65 years, able to speak Bahasa Indonesia, and willing to fill out the informed consent form. We reached 1114 participants through the Google form. We excluded 32 participants because they had duplicate responses when filling out the survey. The final sample size was 1082 participants. 

### 2.2. Procedures

On the first page of the online questionnaire, participants were given an explanation as to the purpose of the survey, the objectives, and that they were agreeing to voluntary participation and consent by completing the questionnaire. In addition, they participated by completing the questionnaire, and in a thank you note at the end, participants were encouraged to invite new respondents from their contact list. Ethical clearance was reviewed and approved by the Institutional Review Board of Institut Ilmu Kesehatan Strada Indonesia (IRB-1911/KEPK/IV/2020). All responses were anonymous and provided with informed consent. There was no monetary compensation for completing the questionnaire.

### 2.3. Measurements

Participants were asked to complete the online sociodemographic questionnaire consisting of information on age, gender, ethnicity, region, marital status, religion, educational level, and health. In terms of participants’ living conditions, extended family included grandparent(s), parent(s) and child(ren) of three or more generations; nuclear family included conventional family of parent(s) and child(ren); and alone defined a Single person household. The physiological impact of anxiety against COVID-19 was assessed using the Depression, Anxiety, and Stress Scale (DASS-21). According to previous studies, a major physiological problem was anxiety, compared to depression and stress [6,16]. Anxiety was assessed by seven items [35]. A higher score indicates a higher level of anxiety. Response options were “never”, “sometimes”, “frequently”, and “every time”, with respective scores of 0, 1, 2, and 3. Cronbach’s α value for the Indonesian version was 0.85 [36].

Participants’ knowledge, attitudes, and practices (KAP) towards COVID-19 were assessed using a KAP questionnaire developed by Zhong et al. (2020) and included statements about clinical presentations, transmission routes, and the prevention and control of COVID-19 (a 12-item scale). The response options were “true”, “false”, and “do not know”; a correct answer was given a score of 1 and an incorrect or “do not know” response was given a score of 0. The total possible knowledge score (K1~K12) ranged from 0–12; a higher score indicates better knowledge of COVID-19. Cronbach’s α value for the KAP-COVID-19 study was 0.71, indicating good internal consistency [21]. Items of attitude in this study were measured by two questions (A1 and A2), including agreement about ultimate control of the disease with three response options, namely agree = 0, disagree, and unknown = 1, and confidence of winning the battle against COVID-19 (agree = 0 and disagree = 1). Moreover, the assessment of participants’ practices consisted of two behavioral questions (P1 and P2), including going to crowded places (yes = 1; no = 0), and wearing a mask when leaving home these days (yes = 0; no = 1). The KAP-COVID-19 questionnaire was developed in China and is available in English. A back-translation method was used to translate the items from English into Bahasa Indonesia and to ensure linguistic and conceptual equivalence using an item discriminant analysis with a *p* value of <0.001.

The Daily Spiritual Experiences Scale (DSES) contains 16 questions, each of which has a 6-point Likert scale ranging from 1 (never or seldom) to 6 (every time). The scale measures ordinary experiences encountered in daily life related to feelings of transcendence [37]. The Indonesian version of the 16-item DSES questionnaire had good internal consistency with Cronbach’s alpha of 0.86 [38]. The more points an individual has, the greater their experience of spirituality.

### 2.4. Statistical Analysis

Descriptive statistics were used to evaluate sociodemographic characteristics, knowledge, attitudes, and practices, additional health information, and spiritual variables between groups, and results are presented as frequencies (*n*) and percentages (%). The percentage of responses was determined according to the total respondents per response for the total question. Continuous variables are presented as the mean and standard deviation (SD) and were evaluated using an independent *t*-test or one-way ANOVA. Absolute values for skewness and kurtosis were used to assess normality of the data; skewness value of 1.779 and kurtosis value of 3.716 indicated a normal distribution [39]. Multicollinearity was calculated using a variance inflation factor (VIF) of <10 [40]. This analysis had a maximum VIF of 2.51, suggesting that the results had low multicollinearity effects. The adjusted beta-coefficients with 95% confidence intervals (CIs) were obtained by performing a multiple linear regression for anxiety related to exposures of interest (spirituality, knowledge, attitudes, and practices) after adjusting for potential confounding variables, including gender, age, ethnicity, region, marital status, religion, educational level, whether the participant was living with family or alone, and the source of health information (family members, professional health education, or online and offline media). SPSS Version 25.0 (Chicago, IL, USA) was used for all statistical analyses, and a *p* value of <0.05 indicated statistically significant.

## 3. Results

Table 1 shows participants’ demographic characteristics. Overall, totals for female participants and non-Javanese participants were 62.1% and 61.7%, respectively; 67.3% of participants had a higher educational level. Most participants were aged 25~39 years (43.9%), were single (51.4%), were Moslem (64.7%), and were from the western part of Indonesia (76.2%). Except for marital status and professional health education information, there were significantly different levels of anxiety in all sociodemographic variables (all *p* < 0.05; Table 1).

The determinants, including KAP and spirituality, of anxiety are presented in Table 2. Interestingly, there were no significant differences between anxiety scores in terms of the following two determinants of KAP: main clinical symptoms of COVID-19 (K1); and residents wearing medical masks to prevent spread of the infection (K8). Levels of anxiety were significantly higher in participants who justified the response (chose ‘correct’) for the following information: unlike the common cold, a stuffy nose, runny nose, and sneezing are less common in persons infected with the COVID-19 virus (K2). Moreover, participants had significantly lower anxiety scores who justified the response (chose ‘correct’) for the following information: currently there is no effective treatment for COVID-19, but early symptomatic and supportive treatment can help most patients recover from the infection (K3); not all persons with COVID-19 will develop severe cases; only those who are elderly, obese, and have chronic illnesses are more likely to be severe cases (K4); the COVID-19 virus can spread via respiratory droplets from infected people (K7); avoiding going to crowded places can prevent the spread of infection (K10); the isolation and treatment of infected people are effective ways to reduce the spread of the virus (K11); and people who have contact with someone infected with the COVID-19 virus should be immediately isolated in a proper place (K12). However, levels of anxiety were significantly lower in participants who justified the response (chose ‘incorrect’) for the information that eating or having contact with wild animals would result in being infected with the COVID-19 virus (K5); persons with COVID-19 cannot spread the virus to others when a fever does not appear (K6); and it is not necessary for children and young adults to take measures to prevent the COVID-19 virus infection (K9). Also, there was a significant association between the total knowledge score of knowledge and level of anxiety due to COVID-19. However, there was not a significant association after adjusting for other covariates. The mean and standard deviation (SD) for anxiety were significantly higher (all *p* values of <0.001) in participants who disagreed that Indonesians will be successful in controlling (A1) and winning the battle against COVID-19 (A2). In analyzing participants’ personal practices, those who reported going to crowded places (P1) and not wearing a mask when outside the home (P2) were significantly correlated with high anxiety scores (both *p* < 0.001). A significantly higher score of anxiety was found in participants with a lower level of spirituality (*p* < 0.001; Table 2).

The adjusted beta-coefficients and 95% CIs of KAP and spirituality for anxiety are presented in Table 3. Three items of knowledge (K3, K4, and K9) were the strongest predictors of the anxiety score, but other items of knowledge (K2, K5, K6, and K7) were not significantly predictors of anxiety score after adjustment for covariates. Participants who justified (chose ‘correct’) that early symptomatic and supportive treatment can help most patients recover from the COVID-19 infection (K3) and people with chronic diseases, who are obese, and who are elderly are more likely to have a possibility of being a severe case (K4) had a significantly lower anxiety score compared to those who responded with ‘incorrect’ after adjusting for covariates. Individuals who justified (chose ‘incorrect’) concerns about the necessity for children and young adults to take measures to prevent the COVID-19 virus infection (K9) had a significantly lower anxiety score compared to those who responded with ‘correct’ after adjusting for covariates. The adjusted beta-coefficients and 95% CIs of the three items of knowledge that predicted the anxiety score were −0.74 (95% CI = −1.47~−0.02), −0.73 (95% CI = −1.43~−0.03), and −0.96 (95% CI = –1.82~−0.09), respectively. Participants who disagreed with the statement that Indonesia could successfully control COVID-19 (A1) and had confidence that Indonesia could win the battle against COVID-19 (A2) had significantly higher anxiety scores (β = 3.23, 95% CI = 2.19~4.26; and β = 2.34, 95% CI = 1.29~3.40, respectively) after adjusting for confounders. Participants with the practice of going to crowded places (P1) had a significantly higher anxiety score (β = 1.23, 95% CI = 0.62~1.83) after controlling for confounding factors. However, there was no significant correlation between the practice of wearing a mask (P2) when leaving the house after controlling for confounding variables. Further analyses revealed that having low spirituality was significantly correlated with a higher anxiety score with an adjusted β of 1.23 (95% CI = 0.65~1.81) among the Indonesian population (Table 3).

## 4. Discussion

This is the first community-based cross-sectional research study with a large sample to determine associations of KAP and spirituality with anxiety among a population during the COVID-19 pandemic in Indonesia. Anxiety symptoms are more likely to occur in the population than in medical professionals and those who have been spending much energy, time, and money on the pandemic [12]. Our findings support an accurate understanding of the source literature related to anxiety among the Indonesian population. In particular, this study revealed that participants who had knowledge and confidence of winning the battle against the disease, agreed with the possibility that the COVID-19 pandemic could be successfully controlled, did not go to crowded places, and had higher spirituality were statistically associated with decreased anxiety in the population. Interestingly, an unexpected result demonstrated that the practice of wearing a mask when leaving home was not significantly correlated with anxiety after adjusting for covariates.

Previous studies revealed that knowledge of the availability and effectiveness of medicines for COVID-19 was negatively correlated with higher anxiety [8]. In a study by Wang et al. [8], children, young adults, those with obesity, those with chronic diseases, and the elderly were also correlated with anxiety. Notably, these findings are in line with the current study: identifying knowledge related to the absence of an effective treatment for COVID-19 and that early symptomatic and supportive treatment will help most patients recover from the infection is positively correlated with a lower level of anxiety. Thus, participants’ knowledge needs to be assessed, especially for children, young adults, those who are obese, those with chronic diseases, and the elderly among the general population, as related to anxiety. A similar study which evaluated knowledge found that most participants had inadequate knowledge and experienced anxiety [41,42]. Good knowledge may help individuals recognize aspects of their emotional experience and learn how to apply emotional regulation and adaptive strategies, particularly for anxiety [43]. Several other studies of hypochondriasis and anxiety disorder suggested that the development and maintenance of health anxiety were subject to selective attention to internal or external health risks [44,45], which may help provide deeper insights into poor knowledge of the disease, and consequently would allow improved psychological regulation control strategies. Our results were inconsistent with other studies, in which no significant correlation between a higher level of knowledge and a low anxiety level was found [11,46,47]. These inconsistent findings might be explained by the fact that participants who seek health information to improve their knowledge of risk factors may play a dominant role, rather than misconceptions about COVID-19 due to rumors, false propaganda, and inaccurate information [46]. These conditions may provoke anxiety, such as panic buying among people during the early phase of the COVID-19 pandemic. Therefore, the specifics of each knowledge question [6,8] or specific psychological knowledge [48] concerning anxiety about COVID-19 need to be re-evaluated during and after the pandemic in terms of psychological problems, including individuals who experienced anxiety during this pandemic period. Consequently, health authorities and health professionals should provide accurate health information through psychological counselling services for stress management, primarily based on evidence regarding knowledge in the general population, to avoid adverse anxiety responses.

In general, having a positive attitude was the most important predictor for a lower level of anxiety [11]. Specifically, our data suggested that individuals with a negative attitude towards confidence that COVID-19 would be successfully controlled, and that Indonesia would win the battle against the disease were independently correlated with higher levels of anxiety. Similar to this study, people who thought that they were unlikely to survive COVID-19 had a 3.9% higher anxiety score [6]. Attitudes agreeing that COVID-19 will be successfully controlled and the battle against it will be won were also effectively correlated with adherence, which further attenuated physiological problems [49,50]. Moreover, several studies revealed that the disadvantages of poor attitudes, such as a high perception of susceptibility and severity, may contribute to higher anxiety problems among the population [11,51,52]. In reality, acute respiratory syndrome resulting from COVID-19 can consequently increase the neutrophil-to-lymphocyte ratio (NLR) in the respiratory tract [53]. Importantly, it was reported that individuals with a more pessimistic attitude regarding their illness were correlated with elevated NLR levels, which might cause more severe symptoms, such as high levels of anxiety [54], leading to a poor quality of life [55]. Therefore, researchers have suggested that those with a positive attitude exhibited declines in inflammation biomarkers, such as NLR [54]. This might subsequently contribute to decreased mental health symptoms, including anxiety [54]. As the pandemic advances and mitigation strategies progress, understanding attitudes is critical among the general population. Importantly, a good attitude is also a key factor in commitment to prevention, as well as decreasing anxiety during this epidemic.

Other evidence of lessons learned from COVID-19 concerns the effects of infectious diseases; this pandemic has altered precautionary practice patterns, which negatively influence anxiety regarding infection among the population [8]. Correspondingly, this study also revealed that the practice of wearing a proper mask when leaving home during the pandemic was not significantly correlated with anxiety. However, avoiding crowded places was significantly correlated with reduced anxiety, and is one of the most important things an individual can do to protect themselves from this infectious pandemic disease. This result is similar to that of a recent review explaining that negative psychological effects during pandemic situations are associated with adverse effects, such as anger, confusion, and stress [56]. In particular, a cross-sectional study of 4700 people in Istanbul, Turkey suggested that the practice of avoiding crowded places had a 0.12-fold lower protective factor against fatigue; however, no significant correlation between wearing a mask to prevent COVID-19 was suggested after adjusting for covariates [57]. Unexpectedly, an inquiry found that wearing masks was not correlated with a low level of anxiety. These inconsistent results might be explained by two possible behaviors among the Indonesian population. First, wearing a mask in public is not a habit among the Indonesian population. The lack of availability of masks indicated that many people could not get them. This was a global problem [58,59]. Second, the Ministry of Health of Indonesia announced that only those with COVID-19 symptoms or relevant diseases should wear medical masks. Uncertainty possibly occurred, created by different recommendations of the Ministry of Health of Indonesia and the Ministry of Health of other countries, such as China, Malaysia, and Vietnam [8,22]. Thus, this issue contributed to various responses when wearing a mask in public areas. In reality, the target of uncovering practices is to modify innate and maladaptive responses, such as fear and anxiety. Anxiety, fear, anger, and a lack of immunity increases pro-inflammatory cytokines and psycho-neuro-immunity against COVID-19. These conditions might provide insights into the pathway that affects how anxiety can increase proinflammatory cytokines [10,60], which further attenuates behaviors or practices [60]. Consequently, health providers or stakeholders should make clear rules related to the use of masks, the time in which to wear them, and the type of mask, in order to ameliorate panic, fear, and confusion, especially for individuals with no access to masks. Strong public support for these practices indicates an opportunity to normalize healthy behaviors and encourage continued use of these and other personal protective behaviors to reduce anxiety as well as mitigate further COVID-19 spread as jurisdictions reopen.

Interestingly, the existence of concerns raised by the COVID-19 pandemic suggests the seriousness of spirituality [30,61]. This finding is in line with a study that explored potential factors affecting anxiety in Spain, in which the authors found that participants with a high score for spirituality had a 0.320-fold lower level of anxiety [33]. Current studies also found that spirituality was correlated with anxiety [62,63]. Spirituality is, indeed, generally helpful for people dealing with major life stressors, as positive psychological concepts including an individual’s core values, deep connections, orientation, and beliefs relate to physical and mental health [64,65]. Additionally, Indonesia has a diverse society with diverse spiritual practices. This unique condition embodies a holistic care approach that recognizes diverse bio-psycho-social-spiritual needs. Therefore, health professionals should develop regulations to achieve holistic mental health services in Indonesia [66]. Conceivably, anxiety is induced by immune system activation and is associated with proinflammatory cytokines, such as interleukin (IL)-6. Increased IL-6 is considered to be correlated with cortisol and could be a risk factor for psychological problems, such as depression and anxiety. Religious psychological concepts also indicate that spirituality mediates the indirect influence of anxiety on IL-6 [64]. The increase in COVID-19-related anxiety cases in Indonesia requires further advocacy of holistic mental health services with spirituality prevention or growth, in which spirituality is recommended among people with anxiety.

When considering anxiety related to the COVID-19 pandemic, providing people with accurate information is the most reasonable prevention against the anxiety. Governments must ascertain the proper propagation of COVID-19 related information. In this pandemic situation, when considering mental health issues due to anxiety, online consultation services might also be more useful in constructing mental health interventions. Finally, these findings and periodic assessments of public KAP, anxiety and spirituality can also advise future planning if subsequent outbreak waves occur, to prevent the dissemination of a new pandemic.

Our study has several strengths. To the best of our knowledge, this is the first paper to estimate associations of spirituality, knowledge, attitude, and practices and their effects on anxiety among the general population in Indonesia. These variables are potentially valuable and might contribute to the recognition and encouragement of strategies for ameliorating anxiety, targeted on encouraging spirituality and KAP. Moreover, a large-scale multisite and cluster-randomized study would provide more-comprehensive evidence regarding individual effects of both KAP and spirituality and other determinants on anxiety in the population that could guide future research implemented in community or clinical settings. 

Along with its strengths, this study also had some limitations. We found that the self-reported score of anxiety by participants might not always be aligned with objective measurements by psychological health professionals. Nonetheless, anxiety, based on personal feelings, is a primary factor during COVID-19 when ensuring the availability of essential preventive and curative healthcare programs [6]. The online assessment approach had a selection bias problem because the Google form was only circulated through social media platforms (WhatsApp, Facebook, Instagram, and Twitter). As a result, there is a possibility that members of the general population without social media may not have been able to access this form. Another limitation is related to the KAP instrument, especially that, regarding attitudes and practices, only two simple questions were used in this study. However, the instrument was adapted from a survey that had been previously tested and used in China [21], Malaysia [22], Jordan, Saudi Arabia, and Kuwait [24]. A further limitation is the lack of participants from the central and easters region, participants living alone, and other sources of information, which future studies could aim to recruit specifically, as this may implicate the generalizability of the findings. However, we adjusted for a considerable number of potential confounding factors by performing a multiple linear regression, thus minimizing the effect of an unequal distribution

## 5. Conclusions

In summary, the present study presents a thorough analysis of Indonesian population’s knowledge, attitudes, and practices regarding anxiety towards the COVID-19 pandemic. These findings indicate crucial roles for health professional educators and practitioners in recognizing and implementing therapeutic approaches, such as improving knowledge, attitudes, and practices to ameliorate anxiety. Hence, it is important to actively monitor the general population’s distress during this pandemic and its aftermath, to detect related protective factors and test for mental health issues, in order to improve policies. Our findings also highlight ensuring the continuity of spirituality during the pandemic.

## Figures and Tables

**Table 1 jcm-09-03798-t001:** Comparisons of participants’ sociodemographic characteristics and anxiety towards the corona virus disease 2019 (COVID-19) pandemic in an Indonesian population (*n* = 1082).

Variables	Total Participants	Anxiety	
	*n* (%)	Mean (SD)	*p*-Value ^a^
Age (years)			
17~24	376 (34.8)	5.13 (5.07)	0.005 ^b^
25~39	475 (43.9)	4.07 (4.37)	
>40	231 (21.3)	4.14 (5.84)	
Gender			
Male	410 (37.9)	5.11 (5.72)	0.001
Female	672 (62.1)	4.05 (4.43)	
Ethnicity			
Javanese	414 (38.3)	3.68 (4.01)	<0.001
Non-Javanese	668 (61.7)	4.93 (5.44)	
Region			
Western part of Indonesia	824 (76.2)	3.58 (3.90)	<0.00 ^b^
Central part of Indonesia	148 (13.7)	7.77 (7.21)	
Eastern part of Indonesia	110 (10.2)	6.55 (5.97)	
Marital status			
Married	526 (48.6)	4.24 (5.22)	0.174
Single	556 (51.4)	4.65 (4.74)	
Religion			
Moslem	700 (64.7)	5.70 (6.15)	<0.001
Non-Moslem	382 (35.3)	3.77 (4.05)	
Education			
ISCED ≥ 3	728 (67.3)	3.54 (4.06)	<0.001
ISCED < 3	354 (32.7)	6.32 (6.07)	
Participants live			
With an extended family	247 (22.8)	5.34 (5.87)	0.006 ^b^
With a nuclear family	811 (75.0)	4.19 (4.66)	
Alone	24 (2.2)	4.08 (4.66)	
Sources of information: Online media			
Yes	329 (30.4)	5.16 (5.51)	0.002
No	753 (69.6)	4.14 (4.70)	
Sources of information: Offline media			
Yes	56 (5.2)	6.61 (5.98)	0.001
No	1026 (94.8)	4.33 (4.89)	
Sources of information: Family members			
Yes	116 (10.7)	5.90 (5.24)	0.001
No	966 (89.3)	4.28 (4.92)	
Sources of information: Professional health			
Yes	119 (11.0)	4.08 (5.26)	0.394
No	963 (89.0)	4.50 (4.94)	

Note: ISCED, International Standard Classification of Education; SD, standard deviation. Data were presented as mean ± SD, frequency and percentage, and *p*-values were calculated using ^a^ independent sample *t*-test, ^b^ one-way ANOVA. A *p*-value of <0.05 indicates statistical significance.

**Table 2 jcm-09-03798-t002:** Comparisons of participant’s knowledge, attitudes, practices, and spirituality with their anxiety scores towards the COVID-19 pandemic among an Indonesian population (*n* = 1082).

Variables	Total Participants	Anxiety	
	*n* (%)	Mean (SD)	*p*-Value ^a^
Knowledge (K) Main clinical symptoms of COVID-19 are a fever, fatigue, dry cough, and myalgia/muscle pain. (K1)			
Incorrect	124 (11.5)	4.52 (4.73)	0.877
Correct	958 (88.5)	4.44 (5.02)	
Unlike the common cold, a stuffy nose, runny nose, and sneezing are less common in persons infected with the COVID-19 virus. (K2)			
Incorrect	325 (30.0)	3.87 (4.09)	0.011
Correct	757 (70.0)	4.70 (5.30)	
Currently, there is no effective treatment for COVID-19, but early symptomatic and supportive treatment can help most patients recover from the infection. (K3)			
Incorrect	189 (17.5)	6.82 (6.74)	<0.001
Correct	893 (82.5)	3.95 (4.36)	
Not all persons with COVID-19 will develop severe disease. Only those who are elderly, obese, and have chronic illnesses are more likely to be severe cases. (K4)			
Incorrect	191 (17.7)	6.04 (6.50)	<0.001
Correct	891 (82.3)	4.11 (4.52)	
Eating or having contact with wild animals could result in being infected with the COVID-19 virus. (K5)			
Correct	662 (61.2)	5.10 (5.42)	<0.001
Incorrect	420 (38.8)	3.42 (3.98)	
Persons with COVID-19 cannot spread the virus to others when a fever does not appear. (K6)			
Correct	281 (26.0)	5.81 (5.75)	<0.001
Incorrect	801 (74.0)	3.98 (4.59)	
The COVID-19 virus spreads via respiratory droplets of infected people. (K7)			
Incorrect	72 (6.7)	6.72 (6.12)	<0.001
Correct	1010 (93.3)	4.29 (4.85)	
Residents can wear medical masks to prevent COVID-19 virus infection. (K8)			
Incorrect	308 (28.5)	4.13 (4.41)	0.181
Correct	774 (71.5)	4.58 (5.18)	
Isolation and treatment of people infected with COVID-19 virus are effective ways to reduce the virus spread. (K11)			
Incorrect	164 (15.2)	7.42 (6.69)	<0.001
Correct	918 (84.8)	3.92 (4.41)	
People who have had contact with someone infected with the COVID-19 virus should immediately be isolated in a proper place. In general, the observation period is 14 days. (K12)			
Incorrect	52 (4.8)	7.43 (6.28)	<0.001
Correct	1030 (95.2)	4.30 (4.86)	
Total score of knowledge			
High (score ≥10)	34 (3.1)	9.12 (7.12)	<0.001
Low (score <9)	1048 (96.9)	4.30 (4.82)	
Attitudes (A) Do you agree that COVID-19 will be successfully controlled? (A1)			
Agree	39 (3.6)	9.90 (7.05)	<0.001
Disagree	1043 (96.4)	4.25 (4.77)	
Do you have any confidence that Indonesia can win the battle against the COVID-19 virus? (A2)			
Agree	680 (62.8)	3.72 (4.72)	<0.001
Disagree	402 (37.2)	5.70 (5.90)	
Practices (P) Recently, have you gone to a crowded place? (P1)			
No	969 (89.6)	3.77 (4.21)	<0.001
Yes	113 (10.4)	10.31 (6.85)	
Recently, have you worn a mask when leaving home? (P2)			
Yes	976 (90.2)	3.84 (4.29)	<0.001
No	106 (9.8)	10.06 (7.03)	
Spirituality			
High (score ≥ 72)	786 (72.6)	3.65 (3.94)	<0.001
Low (score < 72)	296 (27.4)	6.57 (6.58)	

Note: A, attitude; K, knowledge; P, practices; SD, standard deviation. ^a^ Data were presented as mean ± SD, frequency and percentage, and *p*-values were calculated using independent sample *t*-test, A *p*-value of <0.05 indicates statistical significance.

**Table 3 jcm-09-03798-t003:** Adjusted beta-coefficients and 95% confidence intervals (CIs) of knowledge, attitudes, and practices, and anxiety towards the COVID-19 pandemic among the Indonesian population (*n* = 1082).

Variables	Anxiety	
	Unadjusted β-Coef. (95% CI)	Adjusted β-Coef. (95% CI)
Currently, there is no effective treatment for COVID-2019, but early symptomatic and supportive treatment can help most patients recover from the infection. (K3)		
Incorrect	Ref.	Ref.
Correct	−2.86 (−3.63~−2.10) **	−0.74 (−1.47~−0.02) *
Not all persons with COVID-2019 will develop to severe cases. Only those who are elderly, are obese, and have chronic illnesses are more likely to be severe cases. (K4)		
Incorrect	Ref.	Ref.
Correct	−1.93 (−2.70~−1.16) **	−0.73 (−1.43~−0.03) *
It is not necessary for children and young adults to take measures to prevent COVID-19 virus infection. (K9)		
Correct	Ref.	Ref.
Incorrect	−3.49 (−4.30~−2.69) **	−0.96 (−1.82~−0.09) *
Do you agree that COVID-19 will be successfully controlled? (A1)		
Agree	Ref.	Ref.
Disagree	6.55 (5.66~7.44) **	3.23 (2.19~4.26) **
Do you have any confidence that Indonesia can win the battle against the COVID-19 virus? (A2)		
Agree	Ref.	Ref.
Disagree	6.21 (5.29~7.14) **	2.34 (1.29~3.40) **
Recently, have you gone to a crowded place? (P1)		
No	Ref.	Ref.
Yes	2.91 (2.27~3.58) **	1.23 (0.62~1.83) **
Recently, have you worn a mask when leaving home? (P2)		
Yes	Ref.	Ref.
No	3.84 (2.85~4.82) **	0.66 (−0.27~1.59)
Spirituality		
High (score ≥ 72)	Ref.	Ref.
Low (score < 72)	3.08 (2.51~3.65) **	1.23 (0.65~1.81) **

Note: Adjusted beta-coefficients (coef.) and 95% confidence intervals (CIs) were estimated using a multiple linear regression after adjusting for age, gender, ethnicity, region, marital status, religion, educational level, the source of health information, and whether the participants are living with an extended or nuclear family, or alone. * *p* < 0.05; ** *p* < 0.001.

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
