# Peer review of "Effects of Spirituality, Knowledge, Attitudes, and Practices toward Anxiety Regarding COVID-19 among the General Population in INDONESIA: A Cross-Sectional Study"

_jcm, 2020, doi:10.3390/jcm9123798_

Round 1

Reviewer 1 Report

This study designed a cross-sectional study and investigated the association of spirituality, knowledge, attitudes, and practices regarding COVID-19 with anxiety. Corresponding information could be useful in the current pandemic. However, some improvement needs to be done.

  1. It’s unclear how the “Random opportunistic sampling” was done. If it considered the geographic distribution (western, central, and eastern regions).
  2. It’s unclear how the Google Form link was circulated to minimize the selection bias.
  3. Line 149-150. Can’t understand why going to crowed places was set to score 1, but wear mask was set to score 0.
  4. The statistic is not being used correctly in table 1.T-test only suit to compare between two groups. For 3 or more groups, should use the ANOVA test, but not t-test
  5. Table 1 indicated a severe selection bias, but it was not discussed.
  6. Variables listed in table 1 were not explained in the part of the method. For example the definition of extended family and nuclear familiar.
  7. There are overlap or excludability among Sources of information. Don’t know how it was handled, and how it was controlled in table 3.
  8. It unclear why some item presented in table 2 but not shown in table 3, for instance, item K5.
  9. Some mismatch between results and discussion. For instance, Line 276-279, no corresponding results of total knowledge scores were presented.
  10. Results were not discussed sufficiently. For instance, how the information presented in this study can help to improve local corresponding policies, and what the recommendations.
  11. Not all limitations were listed, for instance, the selection bias.

Author Response

RE: jcm-1008062-Version 1

Response to Reviewer 1 Comments

 Dear Reviewer 1,

Thank you for considering our manuscript and for the valuable suggestions, also the opportunity to resubmit a revised manuscript, which help us to improve the article. We carefully revised the manuscript in accordance with your comments. The revised sections of the manuscript are marked with red color. Our point-by-point responses to the comments are as follows. We very much hope the revised manuscript is accepted for publication in Journal of Clinical Medicine. Thank you very much for your consideration.

Point 1. It’s unclear how the “Random opportunistic sampling” was done. If it considered the geographic distribution (western, central, and eastern regions).

Response 1: Thank you for your valuable comments. We have revised the word “random opportunistic sampling” to “convenience sampling” (Please see line 107 on page 3).

Also, let we revise this point to make it clearer and more precise, as follows (Please see line 105-107 on page 3).

“Primary data were collected using a community-based cross-sectional study design to select members of the general population in 17 provinces from western, central, and eastern regions of Indonesia”.

Point 2. It’s unclear how the Google Form link was circulated to minimize the selection bias.

Response 2: Thank you for your valuable suggestion. In this revised manuscript, we reorganized and added a description to make it clear on how the Google Form link was circulated to minimize the selection bias based on the reviewer’s suggestion in the methods section and limitation of our study, as follows:

“Convenience sampling was carried out by distributing an online survey through the Google Form link via WhatsApp, Instagram, Facebook, and Twitter, which are the most popular and accessible social media platforms in Indonesia. While, Instagram and Twitter are more popular among the younger generation, Facebook is generally preferred by older Indonesians. We utilized different approaches to targeting as many respondents as possible all over the region between 7 April and 30 May 2020 data collection period. This involves relying on researchers' technical and personal networks, engaging and sharing the survey through social media influencers and community lenders” (Please see line 107–114 on page 3).

 Online assessment approach had a selection bias problem because the google form only circulated through social media platforms (WhatsApp, Instagram, Facebook, and Twitter). As a result, there is a possibility general population without social media may not have been able to access this form (Please see line 360–363 on page 12).

 Point 3. Line 149-150. Can’t understand why going to crowed places was set to score 1, but wear mask was set to score 0.

Response 3: Thank you for your valuable comments. We would like to confirm that going to crowded places will be increased anxiety. However, wear mask will be decreased anxiety, therefore the values are inverts (Please see line 149–151 on page 4).

Point 4. The statistic is not being used correctly in table 1. T-test only suit to compare between two groups. For 3 or more groups, should use the ANOVA test, but not t-test

Response 4: Thank you for your valuable suggestion. In order to make data to be better presented, we reorganized the table 1 as well as added one-way ANOVA based on reviewer’s suggestion (Please see statistical analysis section; line 166 on page 4; and Table 1; line 184–188 on page 5).

Point 5. Table 1 indicated a severe selection bias, but it was not discussed.

Response 5: Thank you and we appreciate for your valuable comments. In this revised manuscript, we added a description to make a clear the severe selection bias based on the reviewer’s suggestion as follows in the section of the discussion of our study.

“A further limitation is the lack of central region, east region, participants live alone, as well as several source of information, which future study could aim to recruit specifically, as this may implicate the generalizability of the findings. However, we adjusted for a considerable number of potential confounding factors were obtained by performing a multiple linear regression, thus will be minimized the effect of an unequal distribution” (Please see line 366–371 on page 12).

Point 6. Variables listed in table 1 were not explained in the part of the method. For example, the definition of extended family and nuclear familiar.

Response 6: Thank you for your valuable suggestion. In this revised manuscript, we added a description to make a clear the definition of extended family and nuclear familiar based on the reviewer’s suggestion as follows in the section of the method of our study.

“Participants live were used; Extended family included family of grandparent(s), parent(s) and child(ren) [3 or more generation]; Nuclear family included conventional family of parent(s) and child(ren); and Alone defined Single person household” (Please see line 131–133 on page 3).

Point 7. There are overlap or excludability among Sources of information. Don’t know how it was handled, and how it was controlled in table 3.

Response 7: We appreciate this reviewer’s comments. We would like to confirm that our result had a maximum VIF of 2.51, suggesting that the results had low multicollinearity effects, it was handled the overlap information among sources of information (Please see line 168–170 on page 4).

Point 8. It unclear why some item presented in table 2 but not shown in table 3, for instance, item K5.

Response 8:  Thank you very much. We appreciate this reviewer’s comments. In this revised manuscript, we added a description to make a clear the why some item presented in table 2 but not shown in table 3, for instance, item K1, K2, K5 of this study as follows in the section of the result of our study.

“Three items of knowledge (K3, K4, and K9) were the strongest predictors of the anxiety score, but other items of knowledge were not significantly predictors of anxiety score both unadjusted and adjusted beta-coefficients” (Please see line 218–220 on page 6).

Point 9. Some mismatch between results and discussion. For instance, Line 276-279, no corresponding results of total knowledge scores were presented.

Response 9:  Thank you for your valuable comments. To make a clearer and avoiding the mismatch between results and discussion, we deleted sentence "However, in our study, there was no significant association between scores of total knowledge and anxiety levels after adjusting for other covariates. Our results were similar..." (Please line 279 on page 10).

Point 10. Results were not discussed sufficiently. For instance, how the information presented in this study can help to improve local corresponding policies, and what the recommendations.

Response 10: Thank you for your valuable suggestion. In this revised manuscript, we added a description how to the information presented in this study can help to improve local corresponding policies, and what the recommendations based on the reviewer’s suggestion as follows in the section of the conclusion of our study (Please see line 377–380 on page 12).

"Hence it is important to actively monitor the general population's distress during this pandemic and its aftermath to detect related protective factors and test for mental health issues to improve policies”

Point 11. Not all limitations were listed, for instance, the selection bias.

Response 11:  Thank you very much and we appreciate this reviewer’s comments. In this revised manuscript, we added a several limitations such as selection bias based on the reviewer’s suggestion as follows in the section of the discussion (Please see line 360-363 on page 12).

Reviewer 2 Report

Dear authors,

I enjoyed reading your work. However, I have a few serious concerns with the manuscript.

1) highly verbose – the same information with different references is presented heavily. It affects the reading and lowers interest in reading the paper.

For example – look at line 63 from “approximately 8.1% ……… COVID-19 pandemic [6,18,20]” line 69 conveys the same message that is given in line 101 through 103 ending with tumultuous pandemic [6,8,11,14,15].

2) Your title tells the readers very clearly that you are examining the role of spirituality and KAP on anxiety from the COVID-19 pandemic. More than 20% of your sentences either are ending with the COVID-19 pandemic or have this term in it. The overuse of the “COVID-19 pandemic” makes this scientific piece of writing banal. We are not singing a song that needs to be rhyming and should be including words in order to produce good music or sound. It is a scientific paper that attempting to impart a scientific message.

3) what is community-based cross-sectional? What this term (community-based) is adding? Also, how this survey research / cross-sectional research design follows random opportunistic sampling when investigators requested all residents between 17 to 65 years of age to take this survey. Hypothetically, if every person has responded to your survey (which is unrealistic to imagine), had you not included all in your analysis. My comment here again consistent with 1 & 2 in indicating verbiage language.

4) ideally, a value between -1 to +1 for skewness and kurtosis indicates normal distribution. The distribution is not normal if these values are larger. However, this assumption applied very carefully when the sample is larger than 500. In your case, it is 1082. The large skewness and kurtosis values need to be interpreted in light of z-scores in determining if your data follow a normal distribution and meets the assumption of multiple regression.

5) Your recommendation in the conclusion section must be revised. Though we know that people with higher spirituality have lower anxiety in relation to the current pandemic but the suggestion to the health professionals for improving spirituality can indirectly promote religiosity in the community and give a false message that religiosity is a better help than the scientific regimen.  

6) Your first sentence of the abstract -  “The determinants of anxiety and its related factors in the general population affected by the COVID-19 is currently poorly understood.” Could have written as - Currently, the determinants of anxiety and its related factors in the general population affected by the COVID-19 are poorly understood.

Other than the above-mentioned comments, your research has value. Its publication would add knowledge to the scientific body but before this manuscript receives favorable consideration, the language has to be refined.  

Author Response

RE: jcm-1008062-Version 1

Response to Reviewer 2 Comments

Dear Reviewer 2,

Thank you for considering our manuscript and for the valuable suggestions, also the opportunity to resubmit a revised manuscript, which help us to improve the article. We carefully revised the manuscript in accordance with your comments. The revised sections of the manuscript are marked with red color. Our point-by-point responses to the comments are as follows. We very much hope the revised manuscript is accepted for publication in Journal of Clinical Medicine. Thank you very much for your consideration.

Point 1. I enjoyed reading your work.

Response 1: Thank you very much.

Point 2. Highly verbose – the same information with different references is presented heavily. It affects the reading and lowers interest in reading the paper. For example – look at line 63 from “approximately 8.1% ……… COVID-19 pandemic [6,18,20]” line 69 conveys the same message that is given in line 101 through 103 ending with tumultuous pandemic [6,8,11,14,15].

Response 2: Thank you for your valuable comments and suggestion. We re-write the highly verbose sentence of  “COVID-19 occurring unexpectedly and being highly infectious will eventually cause public psychological problems; there is a critical need to understand the public's anxiety and its related factors during this tumultuous pandemic “to be more simplify sentences “Previous studies had showed that COVID-19 cause public psychological problems” (Please see line 98 on page 3).

Point 3. Your title tells the readers very clearly that you are examining the role of spirituality and KAP on anxiety from the COVID-19 pandemic. More than 20% of your sentences either are ending with the COVID-19 pandemic or have this term in it. The overuse of the “COVID-19 pandemic” makes this scientific piece of writing banal. We are not singing a song that needs to be rhyming and should be including words in order to produce good music or sound. It is a scientific paper that attempting to impart a scientific message.

Response 3: Thank you for your valuable suggestion. We appreciate this reviewer’s comment. In this revised manuscript, we revised several sentences based on your comment.

Point 4. What is community-based cross-sectional? What this term (community-based) is adding? Also, how this survey research / cross-sectional research design follows random opportunistic sampling when investigators requested all residents between 17 to 65 years of age to take this survey. Hypothetically, if every person has responded to your survey (which is unrealistic to imagine), had you not included all in your analysis. My comment here again consistent with 1 & 2 in indicating verbiage language.

Response 4: Thank you for your comments and valuable suggestion. Let us clarify and revise this point. “Community-based research takes place in community settings and involves community members in the design to the communities involved. Moreover, community-based is widely used in epidemiology research and was applied in several COVID-19 research (Aylie, N. S., Mekonen, M. A., & Mekuria, R. M. 2020; Devkota, H. R., Sijali, T. R., Bogati, R., Ahmad, M., Shakya, K. L., & Adhikary, P. 2020).        

Also, we have revised the word “random opportunistic sampling” to “convenience sampling” (Please see line 107 on page 3).

 “We utilized different approaches to targeting as many respondents as possible all over the region between 7 April and 30 May 2020 data collection period. This involves relying on researchers' technical and personal networks, engaging and sharing the survey through social media influencers and community lenders” (Please see line 110–114 on page 3).

Point 5. Ideally, a value between -1 to +1 for skewness and kurtosis indicates normal distribution. The distribution is not normal if these values are larger. However, this assumption applied very carefully when the sample is larger than 500. In your case, it is 1082. The large skewness and kurtosis values need to be interpreted in light of z-scores in determining if your data follow a normal distribution and meets the assumption of multiple regression.

Response 5: Thank you for your comments. We would like to confirm that the assessing normal distribution using skewness and kurtosis based on Kim, 2013 revealed that for sample sizes greater than 300, depend on the histograms and the absolute values of skewness and kurtosis without considering z-values. Either an absolute skew value larger than 2 or an absolute kurtosis (proper) larger than 7 may be used as reference values for determining substantial non-normality.

“Absolute values for skewness and kurtosis were used to assess normality of the data; the skewness value of 1.779 and kurtosis value of 3.716 indicated a normal distribution [39].” (Please see line 166-168 on page 4).

References: Kim, H. Y. (2013). Statistical notes for clinical researchers: assessing normal distribution (2) using skewness and kurtosis. Restorative dentistry & endodontics, 38(1), 52-54.

Point 6. Your recommendation in the conclusion section must be revised. Though we know that people with higher spirituality have lower anxiety in relation to the current pandemic but the suggestion to the health professionals for improving spirituality can indirectly promote religiosity in the community and give a false message that religiosity is a better help than the scientific regimen.

Response 6: Thank you for your valuable comments and suggestion. Let we clarify and revise this point to make it clearer and more precise based on the reviewer’s suggestion as follows in the section of the conclusions of our study (Please see line 379–380 on page 12).

“Also, our findings highlight the ensuring the continuity of spirituality during the pandemic.”

Point 7. Your first sentence of the abstract - “The determinants of anxiety and its related factors in the general population affected by the COVID-19 is currently poorly understood.” Could have written as - Currently, the determinants of anxiety and its related factors in the general population affected by the COVID-19 are poorly understood.

Response 7: Thank you for your comments and valuable suggestion. We changed this sentence “The determinants of anxiety and its related factors in the general population affected by the COVID-19 is currently poorly understood” to be “Currently, the determinants of anxiety and its related factors in the general population affected by the COVID-19 are poorly understood” follow reviewer’s suggestion (Please see line 28–29 on page 1).

Point 8. Other than the above-mentioned comments, your research has value. Its publication would add knowledge to the scientific body but before this manuscript receives favorable consideration, the language has to be refined.  

Response 8: Thank you for your valuable suggestion. This manuscript was edited by Taipei Medical University Academic Editing.

Round 2

Reviewer 1 Report

The authors improved the manuscript well, except point 8. the item K5 is significant in table 2, not as what authors's said "not significant predictors of anxiety score both unadjusted and adjusted beta-coefficients". I guess the authors adopted a step-wise regression, so these items were excluded by the fitting process. if so, the authors should clear present it in the part of the method.

Author Response

RE: jcm-1008062-Version 2

Response to Reviewer 1 Comments

Dear Reviewer 1,

Thank you for considering our manuscript and for the valuable suggestions, also the opportunity to resubmit a revised manuscript, which help us to improve the article. We carefully revised the manuscript in accordance with your comments. The revised sections of the manuscript are marked with red color. Our point-by-point responses to the comments are as follows. We very much hope the revised manuscript is accepted for publication in Journal of Clinical Medicine. Thank you very much for your consideration.

Point 1. The authors improved the manuscript well, except point 8. the item K5 is significant in table 2, not as what author’s said "not significant predictors of anxiety score both unadjusted and adjusted beta-coefficients". I guess the authors adopted a step-wise regression, so these items were excluded by the fitting process. if so, the authors should clear present it in the part of the method.

Response 1:  Thank you very much. We appreciate this reviewer’s comments. In this revised manuscript, we added a description to make a clear the why some item presented in table 2 but not shown in table 3, for instance, item K5 of this study as follows in the section of the result of our study.

“The adjusted beta-coefficients and 95% CIs of KAP, and spirituality for anxiety are presented in Table 3. Three items of knowledge (K3, K4, and K9) were the strongest predictors of the anxiety score, but other items of knowledge (K2, K5, K6, and K7) were not significantly predictors of anxiety score after adjustment for covariates” (Please see line 217–220 on page 6).